# A Novel Ensemble Meta-Model for Enhanced Retinal Blood Vessel Segmentation Using Deep Learning Architectures

**DOI:** 10.3390/biomedicines13010141

**Published:** 2025-01-09

**Authors:** Mohamed Chetoui, Moulay A. Akhloufi

**Affiliations:** Perception, Robotics, and Intelligent Machines Lab (PRIME), Department of Computer Science, Université de Moncton, Moncton, NB E1A 3E9, Canada; emc7409@umoncton.ca

**Keywords:** deep learning, blood vessel, U-Net, diabetic retinopathy, image segmentation, medical imaging, ophthalmology

## Abstract

**Background:** Retinal blood vessel segmentation plays an important role in diagnosing retinal diseases such as diabetic retinopathy, glaucoma, and hypertensive retinopathy. Accurate segmentation of blood vessels in retinal images presents a challenging task due to noise, low contrast, and the complex morphology of blood vessel structures. **Methods:** In this study, we propose a novel ensemble learning framework combining four deep learning architectures: U-Net, ResNet50, U-Net with a ResNet50 backbone, and U-Net with a transformer block. Each architecture is customized to enhance feature extraction and segmentation performance. The models are trained on the DRIVE and STARE datasets to improve the degree of generalization and evaluated using the performance metric accuracy, F1-Score, sensitivity, specificity, and AUC. **Results:** The ensemble meta-model integrates predictions from these architectures using a stacking approach, achieving state-of-the-art performance with an accuracy of 0.9778, an AUC of 0.9912, and an F1-Score of 0.8231. These results demonstrate the performance of the proposed technique in identifying thin retinal blood vessels. **Conclusions:** A comparative analysis using qualitative and quantitative results with individual models highlights the robustness of the ensemble framework, especially under conditions of noise and poor visibility.

## 1. Introduction

The retinal blood vessels are crucial for the detection of various diseases, including diabetic retinopathy, occlusion, glaucoma, hypertensive retinopathy, and hemorrhage. Images of the macular region of the eyes give important details for the detection of these conditions [1]. Changes in the retinal blood vessel circulation serve to identify the previously mentioned issues. Diabetic retinopathy (DR) is a condition that, when detected early, can preserve a patient’s vision. In diabetic retinopathy, new blood vessels build in the retina, resulting in damage to the retinal blood circulation. The advanced phases of diabetic retinopathy are defined by damaged blood vessels, neovascularization, and the presence of abnormal blood vessels in the retina [2,3]. The outer layer of the retina grows larger, leading to blood infiltration into the retina, which results in vessel blockage and ocular hemorrhages [4]. DR is classified into four stages: Mild Non-proliferative Retinopathy, Moderate Non-proliferative Retinopathy, Severe Non-proliferative Retinopathy, and Proliferative Retinopathy. Major complications caused by DR include microaneurysms, cotton wool areas, exudates, glaucoma, and retinal detachment [5]. Blood vessel segmentation in retinal images has attracted interest from researchers in recent years due to its remarkable efficacy in the early detection of diabetic retinopathy-related diseases. A minor difference in vascular architecture induces numerous pathological conditions in blood vessels. Abnormal retinal images are essential for diagnosis; however, healthy vascular images are also necessary for optimal comparison. The initial stage in detecting diabetic retinopathy is the extraction of blood vessels from fundus images, requiring retinal blood vessel segmentation. The segmentation method contains various challenges in identifying infected vascular structures during its use. A retinal image may provide details about various objects associated with diseases such as microaneurysm and retinal detachment. Manual extraction of blood vessels requires expert personnel and it is time-consuming.

## 2. Related Work

Object detection includes identifying between foreground and background elements; however, changes in illumination significantly increase detection errors [6]. Manual techniques have been used for the segmentation of retinal fundus images, contributing to the diagnosis of retinal diseases. However, these techniques can require considerable time and effort. In contrast, automated techniques mitigate many of these limitations by providing efficient segmentation, enhancing diagnostic accuracy and significantly reducing the time required for analysis compared to manual methods. Artificial intelligence (AI) has become important in various fields, including healthcare, medicine, agriculture, optical metasurface design [7], dynamic light scattering imaging technologies [8], and virtual content generation [9]. Machine learning and deep learning approaches have demonstrated significant success in medical image analysis, providing advanced methods to enhance diagnostic precision and efficiency.

### 2.1. Traditional and Preprocessing Techniques

Aguirre et al. [10] used the Gabor filter and Gaussian fractional derivative to improve the detection of retinal blood vessel structures and their outlines. The authors used the DRIVE dataset for training and evaluation, and their technique achieved an ACC of 0.9503 and an SN of 0.7854.

### 2.2. U-Net-Based Methods

Sanjeewani et al. [11] presented a U-Net-based architecture to segment retinal blood vessels from fundus images. The authors provided three preprocessing algorithms to enhance the performance of the model. They used the publicly available DRIVE [12] dataset for training and testing their model. The proposed technique achieved 0.9577 for accuracy (ACC), 0.7436 for sensitivity (SE), 0.9838 for specificity (SP), and 0.7931 for F1-Score. Alom et al. [13] proposed a recurrent residual convolutional network based on U-Net for retinal blood vessel segmentation. Their approach achieved an SP of 0.9862 and an AUC of 0.9914. Gegundez et al. [14] introduced a method for segmenting vessels from fundus images from the DRIVE dataset with deep learning. This method employed a deep neural network developed from a simplified U-Net architecture, including residual blocks with batch normalization, achieving an ACC of 0.9575, an SN of 0.8594, and an SP of 0.9706.

### 2.3. Feature-Specific Approaches

Yan et al. [15] presented a three-stage deep learning model to differentiate between thick and thin arteries. A U-Net including atrous convolution (AA-UNet) helped the model identify vessel and non-vessel pixels while leveraging deep features. The model achieved an ACC of 0.9558 when evaluated on the DRIVE test set.

### 2.4. Multi-Scale and Specialized Architectures

Oliveira et al. [16] proposed an innovative method combining multiscale analysis using Stationary Wavelet Transform with a multiscale fully convolutional neural network for retinal vessel segmentation. The authors evaluated their approach on three public datasets, achieving ACC of 0.9576 and 0.9694 and AUC scores of 0.9821 and 0.9905 on the DRIVE and STARE datasets, respectively. Feng et al. [17] introduced a cross-connected convolutional neural network (CCNet) for the automatic segmentation of retinal vascular trees from the DRIVE dataset. Convolutional layers in CCNet extract features and predict pixel classes based on acquired characteristics. The model achieved an ACC of 0.9528 and an SN of 0.7625.

### 2.5. Encoder–Decoder and Thresholding Approaches

Getahun et al. [18] presented a full-scale micro-vessel extraction mechanism based on an encoder–decoder neural network architecture, sigmoid smoothing, and an adaptive threshold method for blood vessel segmentation. The model achieved an AUC of 0.9916 and an ACC of 0.9750 using the STARE dataset. Sathananthavathi et al. [19] employed a region-based technique using a distance regularization term for the segmentation of retinal blood vessels in fundus images. This approach achieved an ACC of 0.9548 using the DRIVE dataset.

In this article, we propose a robust approach for retinal blood vessel segmentation based on modifications to several convolutional network architectures, including U-Net, U-Net with a ResNet50 backbone, an enhanced architecture for U-Net named Conv-Transformer U-Net (CTU-Net) that integrates a transformer block into the U-Net, and a customized ResNet50. The modified architectures are trained using an ensemble learning technique with the stacking method to enhance segmentation performance. We use DRIVE and STARE [13] to train and evaluate the models. We present both qualitative and quantitative results, comparing the individual models’ performance with the ensemble approach, using performance metrics such as Area Under the Curve (AUC), ACC, F1-Score, binary cross-entropy (BCE), SN, and SP.

Our key contributions are as follows:Architectural Modifications: Introducing significant modifications to deep learning models, including integrating a transformer block into U-Net, customizing ResNet50 for segmentation, and enhancing U-Net with a ResNet50 backbone, to improve segmentation accuracy and robustness.Establishing benchmark datasets (DRIVE and STARE) to evaluate and validate the performance of the proposed architectures for blood vessel segmentation and to improve the degree of generalization.Ensemble Learning Framework: Developing a meta-model using a stacking ensemble approach to integrate predictions from the four proposed architectures, resulting in improved overall segmentation performance.

Figure 1 gives an overview of our proposed retinal blood vessel segmentation approach. The input images and masks are processed by different models (U-Net, ResNet50, Conv-Transformer U-Net (CTU-Net), and U-Net with a ResNet50 backbone). The predictions are then combined by a meta-model to produce improved segmentation results.

## 3. Deep Learning Architectures

### 3.1. U-Net Architecture

The U-Net architecture, developed by Ronneberger et al. [20], is a fully convolutional neural network optimized for image segmentation tasks. Originally created to address issues in biological image analysis, it has been applied in different fields, including remote sensing, autonomous driving, and medical imaging. The architecture is characterized by a “U” shape, comprising three primary elements: the encoder part, the bottleneck, and the decoder part. This architecture gives the model the ability to acquire multi-scale contextual information while maintaining spatial accuracy, which is essential for pixel-level prediction tasks.

The encoder gradually reduces the spatial dimensions of the input image while extracting progressively lower- and higher-level characteristics. This is carried out using a sequence of convolutional and pooling layers. Each block in the encoder has two successive 3 × 3 convolutional layers with rectified linear unit (ReLU) activation [21], followed by a 2 × 2 max-pooling layer that reduces the spatial dimensions by half. At each downsampling step, the number of feature channels is increased, allowing the network to identify gradually more complex patterns as it goes deeper. The encoder efficiently reduces the input image into a reduced-dimensional representation while preserving the most significant information.

The bottleneck at the base of the architecture serves as a connection between the encoder and the decoder. The bottleneck extracts the high-level features of the input image. This is achieved by applying convolutional layers with an increased number of filters compared to the preceding layers in the encoder. This layer represents the “reduced” semantic comprehension of the image. The encoder decreases spatial information, while the bottleneck maintains that the model preserves sufficient contextual information for directing subsequent upsampling in the decoder.

The decoder path duplicates the encoder and is tasked with recovering the spatial resolution of the input image while producing segmentation masks. The decoder contains a sequence of transposed convolutional layers, commonly referred to as up-convolutions or deconvolutions, which enhance the resolution of the feature maps. Following each upsampling step, the feature maps from the corresponding encoder layer are concatenated with the upsampled output using skip connections. Skip connections are essential as they recover high-resolution spatial data lost during downsampling, thus providing accurate localization. Following each concatenation, the decoder implements two 3 × 3 convolutional layers with ReLU activation, progressively enhancing the reconstructed feature maps.

The final output layer has a 1 × 1 convolution that decreases the depth of the feature map to correspond with the number of segmentation classes. In binary segmentation tasks, a single output channel is used with a sigmoid activation function, producing probabilities for each pixel. The softmax activation function is used in multi-class segmentation to allocate probability between various categories.

U-Net uses the skip connections. These connections link the encoder and decoder at corresponding levels, allowing the model to use both high-level semantic information from the deeper layers and detailed spatial information from the lower layers. This combination makes U-Net highly efficient for segmentation tasks such as retinal blood vessel segmentation, even with limited training data. Its fully convolutional architecture eliminates the need for fully connected layers, allowing the model to handle input images of varying sizes while significantly reducing the number of trainable parameters. Figure 2 presents the architecture of U-Net.

### 3.2. ResNet50

The ResNet50 architecture, introduced by He et al. [22], is a deep convolutional neural network optimized for image classification applications. Its primary novelty is the implementation of residual connections, which reduce the vanishing gradient issue and allow the training of very deep networks. ResNet50 demonstrates adaptation and robustness, making it a superior backbone for image segmentation tasks when combined with an appropriate decoder. A modified ResNet50 for segmentation maintains its basic residual architecture while adding further elements for complete pixel-wise predictions.

The encoder of the modified ResNet50 serves as a feature extractor, producing complex hierarchical representations of the input image. The architecture includes an initial convolutional layer including a 7 × 7 kernel and a stride of 2, followed by a max-pooling layer. These layers aim to reduce the spatial dimensions of the input image, extracting essential information. The network consists of four stages of residual blocks, each containing convolutional layers, batch normalization, and ReLU activations. Residual blocks contain shortcut connections that skip one or more layers, providing efficient gradient propagation and maintaining low-level information. ResNet50’s depth allows the extraction of high-level semantic features necessary for segmentation tasks. Pre-trained weights on large data sets such as ImageNet [23] are frequently employed to initialize the encoder, providing a robust basis for transfer learning.

In our proposed modification of the architectures, the decoder uses pixels to upsample the encoded features to the original input resolution, providing pixel-level segmentation maps. The decoder uses transposed convolutions and bilinear upsampling techniques to incrementally recover spatial dimensions. Skip connections are used between the encoder and decoder phases to recover spatial information that is decreased during downsampling. These connections integrate the first layers of ResNet50 with the corresponding decoder layers, combining low-level spatial variables with high-level semantic features. This combination provides precise boundary segmentation of retinal blood vessels.

A major change in the architecture includes the addition of a pyramid pooling module which improves the encoder’s performance to capture multi-scale contextual information. These modules implement convolutions with various dilation rates, enabling the network to collect features at variable receptive field dimensions.

The output layer of the modified ResNet50 for retinal blood vessel segmentation consists of a 1 × 1 convolutional layer that reduces the feature map depth to correspond with the number of segmentation classes (blood vessel). Segmentation includes a single output channel with a sigmoid activation function to generate pixel-wise probability. The architecture allows both fine-grained and coarse segmentation tasks, making it suitable for retinal blood vessel segmentation.

The improved ResNet50’s residual connections and hierarchical feature extraction performance make it efficient for the segmentation of blood vessels. Figure 3 gives an overview of the modified ResNet50.

### 3.3. U-Net with ResNet50 Backbone

The implementation of a ResNet50 backbone into the U-Net architecture provides a model that combines the best features of both architectures, offering efficient stability between deep feature extraction and accurate spatial localization for retinal blood vessel segmentation. This modification combines U-Net’s standard encoder with ResNet’s residual architecture, allowing the model to use pre-trained weights and hierarchical feature representations while maintaining the basic framework of the U-Net decoder.

The primary advantage of this method is the connection between the ResNet50 encoder and the skip connections of U-Net. The encoder’s residual blocks gradually extract detailed features, while the skip connections maintain fine-grained data from previous layers. These connections reduce the loss of spatial information, allowing the decoder to recreate segmentation masks with high precision. The ResNet50 backbone, compared to a conventional U-Net, allows deeper and more robust feature extraction, making the model especially proficient for tasks characterized by high semantic complexity.

The proposed architectures includes multi-scale feature integration using feature maps from various ResNet50 stages. These multi-scale features improve the model’s flexibility in retinal blood vessel segmentation, including objects of different sizes and with complex borders. By adopting the decoder’s upsampling architecture, the model achieves a balance between computational efficiency and segmentation precision, without requiring substantial modifications to the U-Net framework.

This hybrid architecture reduces the limitations of both U-Net and ResNet50. It augments U-Net’s dependence on local features by integrating ResNet50’s global feature representation. The result is a diverse and high-performing model suitable for situations where accuracy, scalability, and efficiency are crucial for blood vessel segmentation. Figure 4 gives a diagram of our proposed U-Net with a ResNet50 backbone.

### 3.4. Conv-Transformer U-Net (CTU-Net)

Our paper presents a modification of the U-Net architecture for the segmentation of blood vessels in medical imaging. To solve the problems of traditional convolutional methods for identifying long-range dependencies, we integrate a Transformer block [24] into the U-Net architecture, particularly during the bottleneck stage. This modification improves the capacity of the model to include global context, crucial for the accurate identification of complex, long structures such as blood vessels. Blood vessel segmentation is hard due to the complex and small structures that frequently extend over a large area. Traditional U-Net structures, efficient at preserving spatial information via skip connections, mostly rely on local convolutional filters. This limits their ability to represent the global relationships between vessel segments that are directly separated though semantically connected. Our architecture reduces this issue by incorporating a Transformer block that uses the self-attention technique to collect dependencies throughout every feature space. In our modified U-Net, the encoder produces hierarchical features using convolutional layers, reducing spatial resolution but enhancing semantic abstraction. At the bottleneck, we replace the standard convolutional layers with a Transformer block. The feature maps are initially flattened into a sequence of patches, that serve as input tokens for the Transformer. Each token communicates with all other tokens through multi-head self-attention layers, allowing the model to learn global context and long-range dependencies. This is particularly efficient for blood vessel segmentation, where the continuation and connection of blood vessel structures throughout the image are crucial.

The self-attention process is enhanced by feed-forward layers in the Transformer block, which improve the feature representations. The processed tokens are converted into a spatial feature map, preserving the necessary resolution for decoding. The decoder uses skip connections to merge high-resolution spatial details from the encoder with the globally enriched features from the Transformer-augmented bottleneck. This combination ensures the keeping of both local characteristics and global vessel connection in the final segmentation result. Figure 5 presents the proposed CTU-Net architecture.

## 4. Experimental Results

### 4.1. Dataset

DRIVE [12] and STARE [13] retinal image datasets were used in this study for training and evaluating the proposed approach. Most retinal blood vessel segmentation methods in the literature use these datasets. They contain manually performed segmentation provided for each image by blood vessel structure specialists. This helps in the training of networks through the automatic extraction of vessel and non-vessel samples from images, the evaluation of these images against a reference gold standard vasculature image, and comparison with past published techniques.

The DRIVE (Digital Retinal Images for vessel Extraction) dataset contains 40 RGB retinal images, obtained using a non-mydriatic 3CCD camera with a 45-degree field of view (FOV). The images possess a resolution of 768 × 584 pixels and a circular field of view of roughly 540 pixels. The dataset is categorized into two groups of 20 images each: the training set and the testing set. Figure 6 gives an example of images and corresponding masks from the DRIVE dataset.

The STARE (STructured Analysis of the Retina) dataset consists of 20 color images of the eye fundus captured with a TopCon TRV-50 camera with a 35° field of view and digitized at 700 × 605 pixels (about 650 pixels in retinal diameter). Each image was manually annotated by two human observers. Figure 7 gives an example of images and corresponding masks from the STARE dataset.

For our experiments, we divided the images into training and testing sets. Specifically, we allocated 20 images from the DRIVE dataset for training and reserved the remaining 20 for testing. In the case of the STARE dataset, we allocated 14 images for training and used the remaining 6 for testing. To improve generalization and robustness, data augmentation techniques were used, including random rotations (up to 15°), width and height shifts (10%), zooming (up to 20%), and horizontal flipping. The images were resized to 256 × 256 for all models.

### 4.2. Metrics

For training and evaluation of the proposed models for retinal blood vessel segmentation, we used the following performance metrics: accuracy, AUC, sensitivity, specificity, F1-Score, and BCE.

Accuracy measures the proportion of correct predictions among all predictions made by the model. It is defined as:$$\mathrm{Accuracy}=\frac{\mathrm{TP}+\mathrm{TN}}{\mathrm{TP}+\mathrm{TN}+\mathrm{FP}+\mathrm{FN}}$$
where TP, TN, FP, and FN are the numbers of true positives, true negatives, false positives, and false negatives, respectively.

Sensitivity measures the model’s performance to correctly identify positive cases. It is expressed as$$\mathrm{Sensitivity}=\frac{\mathrm{TP}}{\mathrm{TP}+\mathrm{FN}}$$

Specificity evaluates the model’s performance to correctly identify negative cases. It is defined as$$\mathrm{Specificity}=\frac{\mathrm{TN}}{\mathrm{TN}+\mathrm{FP}}$$

The F1-Score is a means of measuring precision and recall, offering a single metric that balances the two. It is calculated as$$\text{F1-Score}=2\cdot \frac{\mathrm{Precision}\cdot \mathrm{Sensitivity}}{\mathrm{Precision}+\mathrm{Sensitivity}}$$

Binary cross-entropy is a loss function used in our study to measure the divergence between predicted probabilities and true labels. It is defined as$$\mathrm{BCE}=-\frac{1}{N}\sum_{i=1}^{N}\left[y_{i}\log\left({\hat{y}}_{i}\right)+\left(1-y_{i}\right)\log\left(1-{\hat{y}}_{i}\right)\right]$$
where $y_{i}$ is the ground truth label for the *i* sample, ${\hat{y}}_{i}$ is the predicted probability, and *N* is the total number of samples.

### 4.3. Training

Model training and model testing in this paper’s experiments were conducted on an NVidia V100SXM2 (16G memory) and an AMD EPYC 7532 (Zen 2) 2.40 GHz CPU.

All proposed deep learning models were developed using the Keras library, compiled with an Adam optimizer [25] with a learning rate of 0.01. The training process used 500 epochs to allow sufficient time for best convergence, particularly for the Trans-U-Net. Each epoch included a number of steps determined by dividing the training dataset size by a batch size of 32. Validation was performed at the end of every epoch with a similar configuration. A ModelCheckpoint callback was used to save the best-performing model based on the validation loss, ensuring that the most accurate version of the model was retained. The four best models for the four architectures were used to perform the ensemble learning training.

### 4.4. Results

Figure 8 represents the training and validation curves for the U-Net model with the best score for each metric. As we can see, the training BCE loss decreases consistently over 500 epochs, showing that the model efficiently learns from the training dataset. The validation BCE initially follows a similar training curve and achieves a score of 0.0909. The F1-Score curve, which balances precision and recall in the training phase, increases over 500 epochs, demonstrating the model’s improved balance between detecting blood vessel pixels (recall) and avoiding false positives (precision). Figure 9 shows the training and validation curves for the customized ResNet50 model. As we can see, the model obtains a very close score to U-Net in terms of BCE, but as the validation curve appears noisier, it also decreases, which shows that the model is able to generalize, but with less stability compared to the training data. The remaining scores appear almost the same compared to the U-Net model. Figure 10 presents the CTU-Net training and validation curves, and the model obtains a high accuracy score compared to U-Net and ResNet50 architectures; however, it obtains very close validation scores in terms of AUC and F1-Score compared to the U-Net and ResNet50 models. Figure 11 shows training and validation curves of U-Net with a ResNet50 backbone, and we can see that the curve and the score of BCE are similar to those of the U-Net and ResNet50 architectures, but lower than those of CTU-Net in terms of F1-Score. All the proposed models obtained scores ranging from 0.9663 to 0.9690.

The four proposed architectures were used for the segmentation of retinal blood vessels, with each contributing differently to the process. These models were designed to extract important features for segmenting retinal blood vessels.

Ensemble learning has been widely used in previous studies to improve performance in various segmentation tasks, including retinal blood vessel segmentation. Inspired by these findings, we propose an ensemble learning approach that combines the performance of the four architectures. The aim is to develop a robust meta-model that enhances segmentation accuracy and precision. The details of the proposed meta-model are provided in the following subsection.

## 5. Meta-Model

In our study, we use a stacking [26] ensemble learning approach to enhance the segmentation of retinal blood vessels in fundus retina images. Stacking is a supervised learning technique that integrates the predictions of the four customized architectures to produce a more precise and robust result. In contrast with traditional ensemble methods like bagging or boosting, stacking employs a meta-model to efficiently combine the outputs of various base models.

For the input image *x*, each of the four models (U-Net, ResNet50, U-Net with a ResNet50 backbone, and CTU-Net) generates a segmentation mask. The masks, denoted as $P_{1}\left(x\right)$, $P_{2}\left(x\right)$, $P_{3}\left(x\right)$, and $P_{4}\left(x\right)$, contain various spatial and hierarchical characteristics of the retinal blood vascular architectures. The predictions from the models are combined to create a feature tensor Z(x):(1)$$Z\left(x\right)=\left[\begin{array}{c} P_{1}\left(x\right)\\ P_{2}\left(x\right)\\ P_{3}\left(x\right)\\ P_{4}\left(x\right) \end{array}\right]$$

The combined representation Z(x) serves as the input for a meta-model *g*. The meta-model is trained to reduce the differences between the ground truth mask *y* and the final prediction $P_{\mathrm{final}}\left(x\right)$, calculated as(2)$$P_{\mathrm{final}}\left(x\right)=g\left(Z\left(x\right)\right)$$

The meta-model determines how to distribute the appropriate weights to the outputs of the base models to optimize performance. It combines their complementary features, including U-Net’s efficiency, with preserving complex information, ResNet50’s hierarchical feature extraction, and the transformer block’s performance for representing long-range dependencies. This method reduces false positives and false negatives in complex regions in the retina, such as vessel connections, which are frequent challenges in blood vessel segmentation. To train the meta-model, we use pixel-wise binary cross-entropy loss, defined as(3)$$L\left(y,P_{\mathrm{final}}\left(x\right)\right)=-\frac{1}{N}\sum_{i=1}^{N}\left[y_{i}\log P_{\mathrm{final},i}\left(x\right)+\left(1-y_{i}\right)\log\left(1-P_{\mathrm{final},i}\left(x\right)\right)\right]$$
where $y_{i}$ is the ground truth label for pixel *i*, $P_{\mathrm{final},i}\left(x\right)$ is the predicted probability for pixel *i*, and *N* is the total number of pixels.

The proposed meta-model is presented in Figure 12. The model input consists of concatenated outputs from four models. The architecture includes two convolutional layers: one with 64 filters and a kernel size of 3×3, followed by a second convolutional layer with 1 filter and a kernel size of 1×1. The final output is a single-class segmentation mask. We kept the same hyperparameters for training the meta-model. Figure 13, Figure 14 and Figure 15 present the loss, accuracy, and F1-Score curves for training the meta-model over 500 epochs. As we can see, the scores clearly improved compared to individual models’ training and validation. We obtained an accuracy score of 0.9740 which is higher compared to the value of 0.9690 obtained by CTU-Net. The BCE loss decreased to 0.006 and the F1-Score increased to 0.8260.

The ablation study presented in Table 1 evaluates ten combinations of models, testing the performance of two-model, three-model, and four-model ensembles for blood vessel segmentation. The metrics analyzed include loss, accuracy (Acc), sensitivity (SN), specificity (SP), Area Under the Curve (AUC), and F1-Score, providing a comprehensive evaluation of each experiment. Among the two-model combinations, U-Net + ResNet50 achieved a loss of 0.0730 and an F1-Score of 0.8130, indicating good performance in terms of balanced precision and recall. However, its sensitivity (SN = 0.7787) is moderate compared to other combinations, suggesting some limitations in accurately segmenting blood vessels. Similarly, ResNet50 + CTU-Net performs worse, with marginally higher loss (0.0757) and a lower F1-Score (0.8093), despite achieving higher sensitivity (SN = 0.7821). In the three-model experiments, ResNet50 + CTU-Net + U-Net-ResNet50-backbone provided a good F1-Score (0.8172), AUC (0.9850), and sensitivity (SN = 0.7838). This suggests that including multiple architectures enhances the model’s ability to generalize and segment blood vessels accurately. Another interesting result is that of ResNet50 + U-Net + U-Net-ResNet50-backbone, which also demonstrates high performance, with an F1-Score of 0.8165, competitive sensitivity (SN = 0.7836), and low loss (0.0698). However, the most notable results were observed in the four-model combination (U-Net + ResNet50 + CTU-Net + U-Net-ResNet50-backbone), which achieved the highest metrics across the board. This test provided an F1-Score of 0.8231, an AUC of 0.9912, the lowest loss of 0.0591, high specificity (SP = 0.9920), and competitive sensitivity (SN = 0.7790). These results demonstrate the importance of integrating all models simultaneously. By leveraging the complementary strengths of U-Net, ResNet50, CTU-Net, and U-Net-ResNet50-backbone, the four-model combination maximized detection accuracy, minimized false positives, and improved overall robustness.

Table 2 and Table 3 compare the performance of various models in the task of retinal blood vessel segmentation, evaluated on the DRIVE and STARE datasets. The meta-model demonstrates superior results across both datasets.

On the DRIVE dataset, the meta-model achieves the lowest loss (0.0591) and the highest accuracy (0.9778), showing its ability to accurately segment retinal blood vessels. It also obtains the highest specificity (0.9920), which indicates its robustness in correctly identifying non-vessel pixels while minimizing false positives. Furthermore, the meta-model achieves the highest AUC (0.9912), reflecting its efficiency in distinguishing between vessel and non-vessel regions, and the highest F1-Score (0.8231), showing a strong balance between precision and recall. Although ResNet50 achieves slightly higher sensitivity (0.8039), making it better at detecting vessel pixels, the meta-model’s sensitivity (0.7790) remains competitive and it excels in other metrics.

On the STARE dataset, the meta-model also outperforms the other models in most metrics, achieving the lowest loss (0.0649) and the highest accuracy (0.9760), which further confirms its performance in retinal blood vessel segmentation. It achieves a specificity of 0.9912, only marginally lower than U-Net-ResNet50-backbone (0.9914), and the highest AUC (0.9855), demonstrating its ability to handle complex vessel structures. Additionally, the meta-model records the highest F1-Score (0.8082), reflecting its better performance in both precision and recall. Although CTU-Net achieves the highest sensitivity (0.8497), the meta-Model’s sensitivity (0.7630) provides a good balance with its other superior metrics. This shows the robustness of our proposed approach for segmenting retinal blood vessels.

After training and testing these models, we evaluated their robustness on test images to examine their qualitative performance in blood vessel segmentation. Figure 16 shows the results of the meta-model, which combines the outputs of multiple individual models. Figure 16a presents the original retinal images, Figure 16b shows the ground truth (manually labeled vessels), and Figure 16c presents the predictions from the meta-model. The meta-model closely matches the ground truth, capturing both main vessels and finer branches with good continuity across all images. Figure 17 compares the predictions of individual models. Figure 17a contains the original images, while Figure 17b–e show the results from U-Net, ResNet50, CTU-Net, and U-Net with a ResNet50 backbone, respectively. U-Net performs efficiently on larger vessels but faces challenges with finer details. ResNet50 focuses on broader vessels, and CTU-Net detects small vessel more effectively, but it introduces noise. The U-Net with a ResNet50 backbone balances large- and small-vessel detection. The meta-model outperforms the individual models by combining their performance. It segments both large and small vessels more accurately, reduces false positives, and improves vessel structure integration. This demonstrates the robustness of ensemble learning for retinal blood vessel segmentation. Our approach demonstrates interesting performance results compared to several recent studies using the DRIVE and STARE datasets, as shown in Table 4 and Table 5.

## 6. Discussion

The proposed approach improves retinal blood vessel segmentation by combining customized deep learning architectures, including U-Net, ResNet50, U-Net with a ResNet50 backbone, and CTU-Net, integrated through a stacking ensemble learning approach. The results of the meta-model notably enhanced the segmentation accuracy, achieving an interesting score of 0.9778 in accuracy and a F1-Score of 0.8231, outperforming the individual architectures. The integration of diverse architectures highlighted the unique contributions of each model:U-Net: Efficient in capturing small spatial details, particularly in regions with high vessel density.ResNet50: Robust in segmenting major vessel structures with minimal noise due to its hierarchical feature extraction performance.CTU-Net: Demonstrates improved global context modeling, and provides better detection for long and thin vessels.U-Net with a ResNet50 backbone: Balanced performance, and performs particularly well in challenging regions like vessel crossings.

The stacking ensemble approach effectively combined the performance of these modified architectures, addressing their individual limitations. By including a meta-model, the ensembling reduced segmentation errors. The meta-model effectively combined the performance of segmentation from different architectures, demonstrating robustness for retinal blood vessel segmentation.

Despite the improvements, the meta-model has certain limitations, including computational complexity. The ensemble approach increases computational requirements due to the simultaneous training and evaluation of multiple architectures. To resolve this issue, future works may include developing model compression techniques, such as quantization, knowledge distillation, and pruning, to reduce the size and computational load of the networks while maintaining their performance. In addition, although the meta-model improves the segmentation for thin vessels, challenges persist in consistently identifying regions with extreme noise or poor visibility. Denoising approaches can be explored to help with this issue.

When compared with existing studies, the individual and ensemble models exhibit superior performance metrics, including higher accuracy and specificity. The integration of transformer blocks in the CTU-Net architecture and the ensemble strategy outperformed conventional methods in capturing both local and global vessel features.

## 7. Conclusions

This study presents a robust approach to retinal blood vessel segmentation by leveraging advanced modifications to deep learning architectures and an ensemble learning framework. The integration of a transformer block in CTU-Net, the use of a ResNet50 backbone in U-Net, and the customization of ResNet50 for segmentation tasks collectively contribute to improved feature extraction, global context modeling, and segmentation precision. By combining these architectures through a stacking ensemble learning technique, the proposed meta-model demonstrates superior performance, achieving state-of-the-art results, including a BCE loss of 0.0591, accuracy of 0.9778, sensitivity of 0.7790, specificity of 0.9920, an AUC of 0.9895, and an F1-Score of 0.8231 on the DRIVE dataset. Interesting results were obtained on the STARE dataset, with an AUC 0.9855 and a specificity of 0.9912. These results highlight the efficiency of the proposed ensemble learning strategy in addressing challenges such as the segmentation of thin and complex vessel structures.

Future research directions include optimizing the framework for real-time applications, expanding its performance to other imaging domains, and exploring the integration of additional datasets to further enhance its generalizability.

## Figures and Tables

**Figure 1 biomedicines-13-00141-f001:**
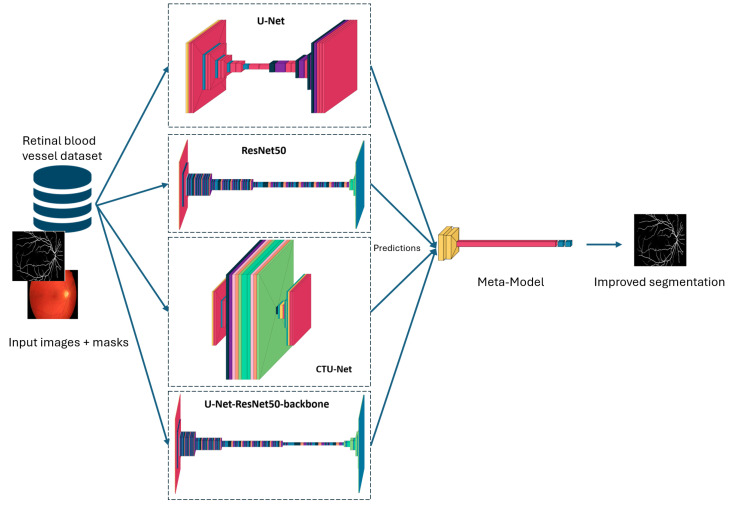
Proposed methodology for retinal blood vessel segmentation.

**Figure 2 biomedicines-13-00141-f002:**
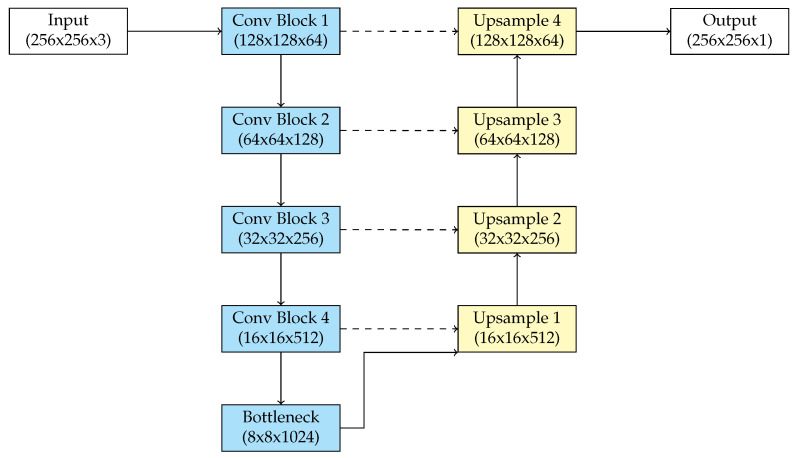
U-Net architecture.

**Figure 3 biomedicines-13-00141-f003:**
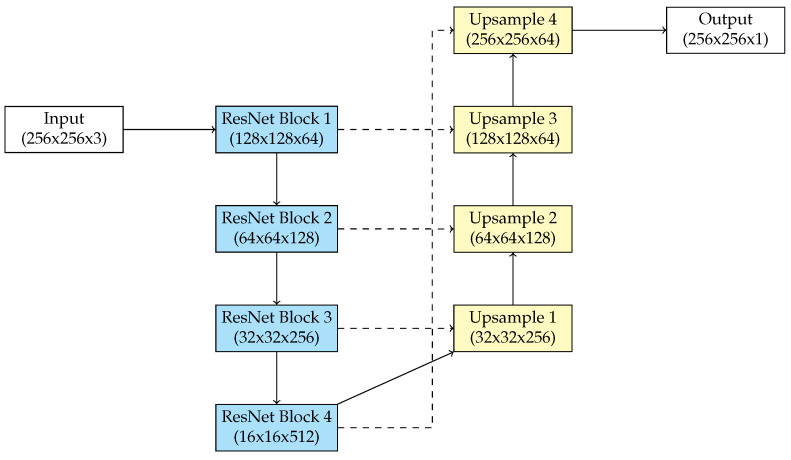
The modified ResNet50.

**Figure 4 biomedicines-13-00141-f004:**
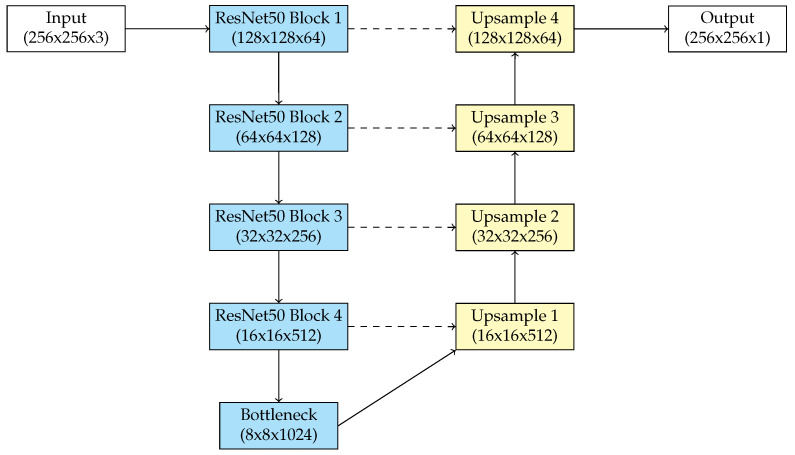
Diagram of U-Net with ResNet50 backbone.

**Figure 5 biomedicines-13-00141-f005:**
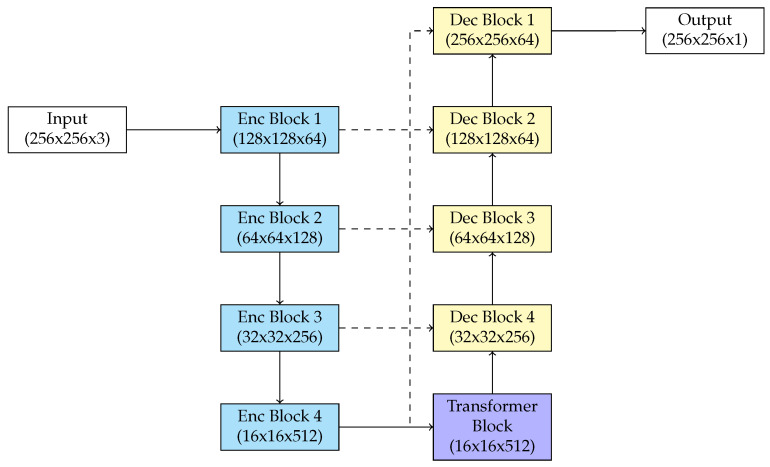
Diagram of proposed CTU-Net.

**Figure 6 biomedicines-13-00141-f006:**
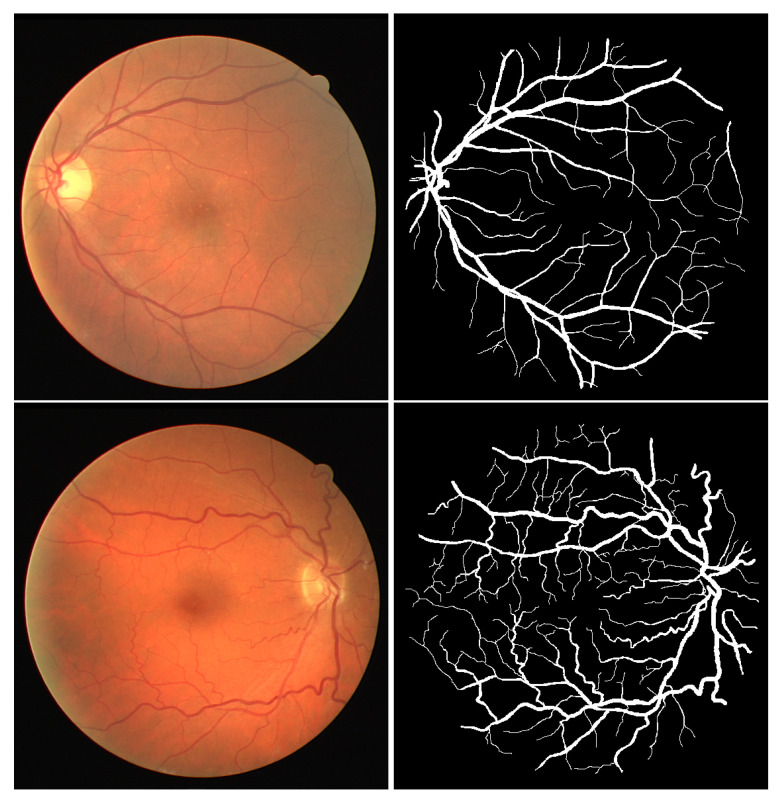
Example of images with corresponding masks from DRIVE dataset [12].

**Figure 7 biomedicines-13-00141-f007:**
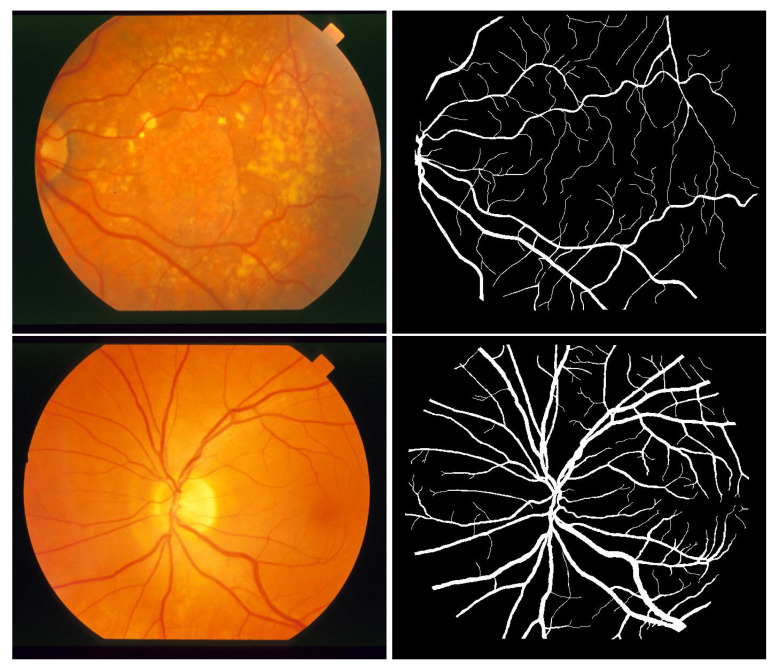
Example of images corresponding masks from STARE dataset [13].

**Figure 8 biomedicines-13-00141-f008:**
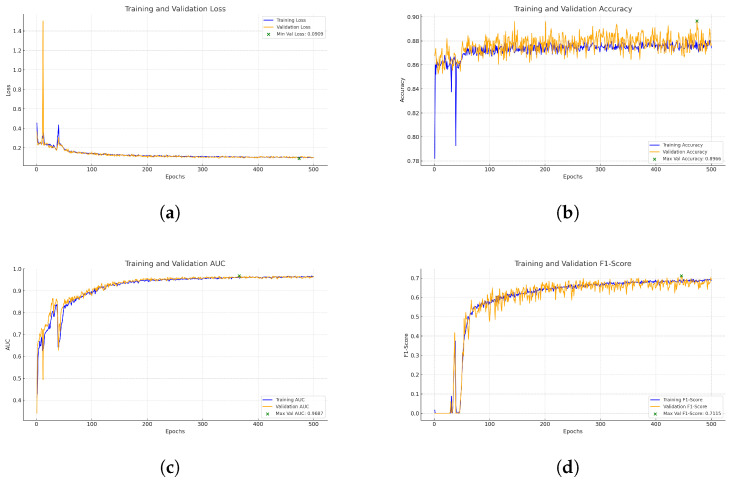
Performance metrics for U-Net training and validation. Each subfigure illustrates a key metric tracked during training: (**a**) BCE loss, (**b**) accuracy, (**c**) AUC, and (**d**) F1-Score.

**Figure 9 biomedicines-13-00141-f009:**
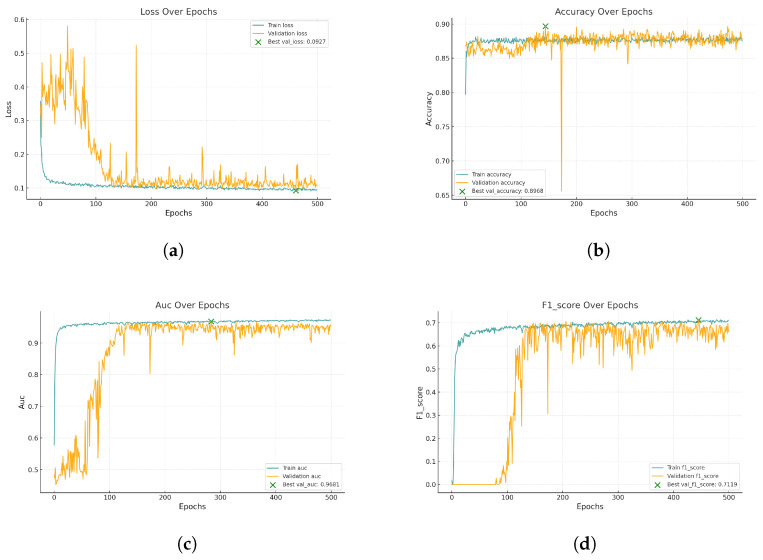
Performance metrics for modified ResNet50 model training and validation. Each subfigure illustrates a key metric tracked during training: (**a**) BCE loss, (**b**) accuracy, (**c**) AUC, and (**d**) F1-Score.

**Figure 10 biomedicines-13-00141-f010:**
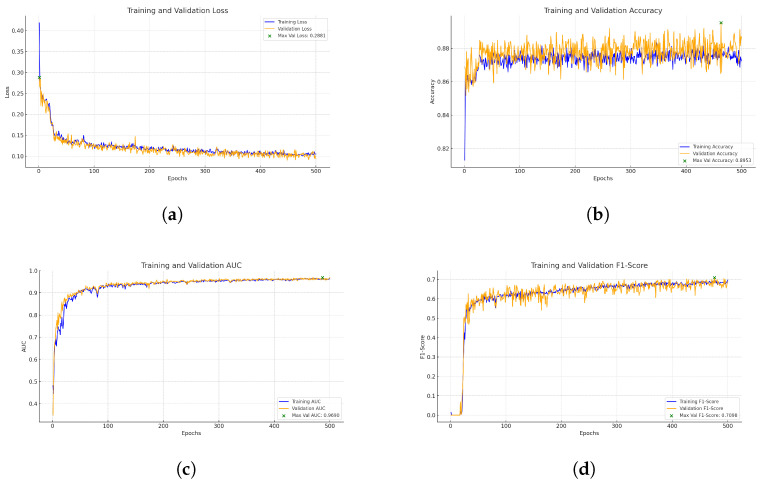
Performance metrics for CTU-Net model training and validation. Each subfigure illustrates a key metric tracked during training: (**a**) BCE loss, (**b**) Accuracy, (**c**) AUC, and (**d**) F1-Score.

**Figure 11 biomedicines-13-00141-f011:**
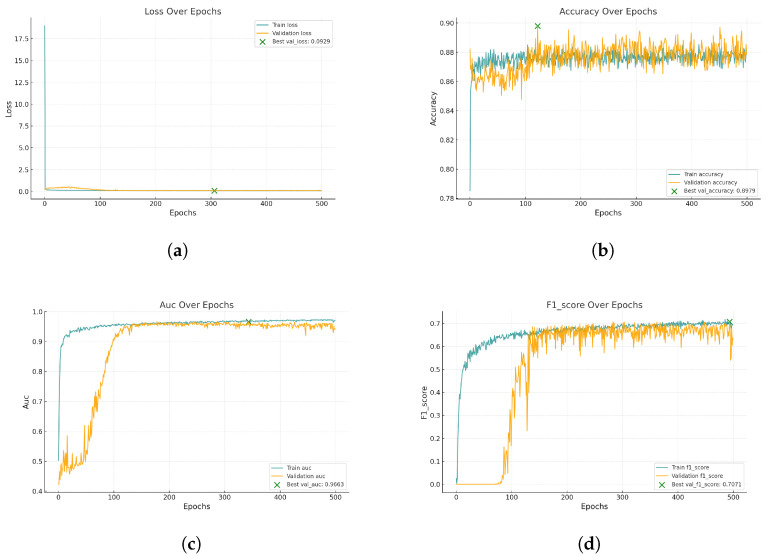
Performance metrics for U-Net with the ResNet50 backbone in model training and validation. Each subfigure illustrates a key metric tracked during training: (**a**) BCE loss, (**b**) accuracy, (**c**) AUC, and (**d**) F1-Score.

**Figure 12 biomedicines-13-00141-f012:**
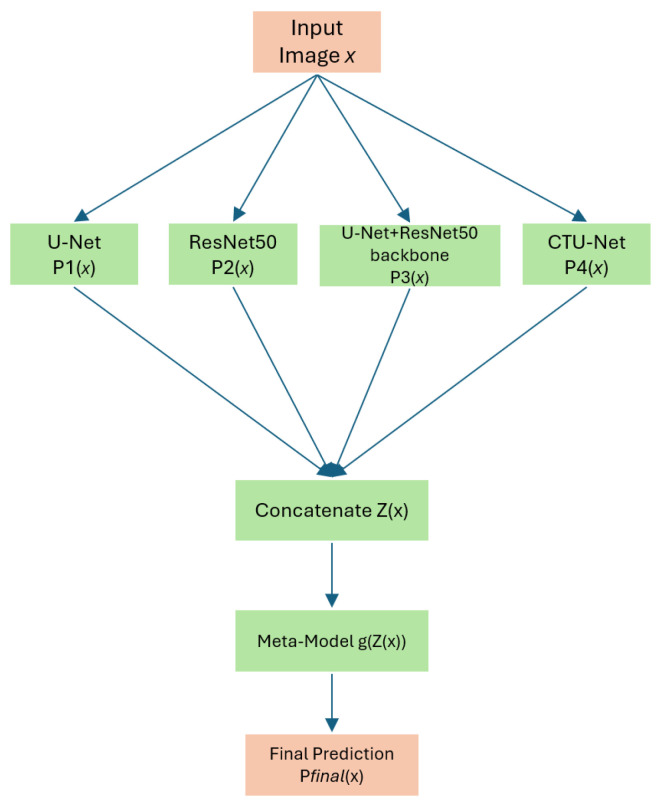
A diagram of the meta-model for stacking-based blood vessel segmentation. The meta-model combines predictions from four base models, U-Net, ResNet50, U-Net with a ResNet50 backbone, and CTU-Net with a transformer block, to produce a final segmentation mask.

**Figure 13 biomedicines-13-00141-f013:**
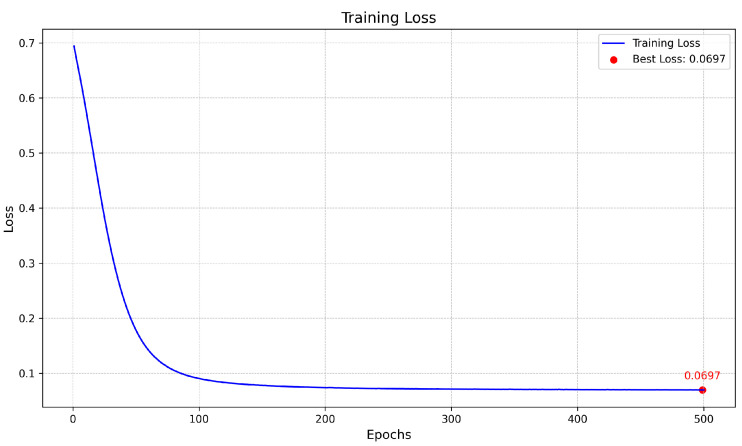
Meta-model training BCE loss over 500 epochs with best value annotated.

**Figure 14 biomedicines-13-00141-f014:**
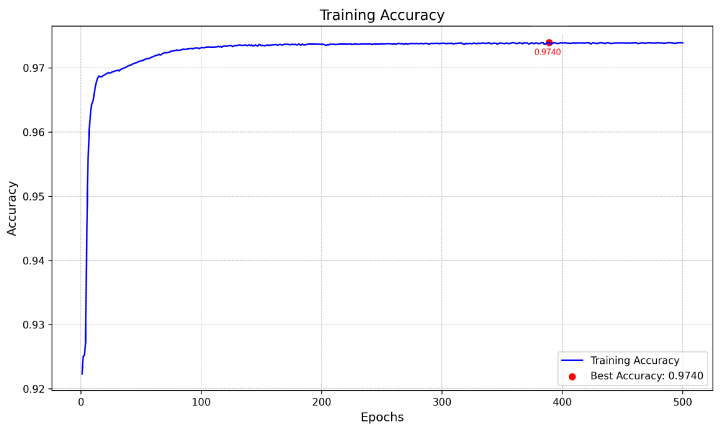
Meta-model training accuracy over 500 epochs with best value annotated.

**Figure 15 biomedicines-13-00141-f015:**
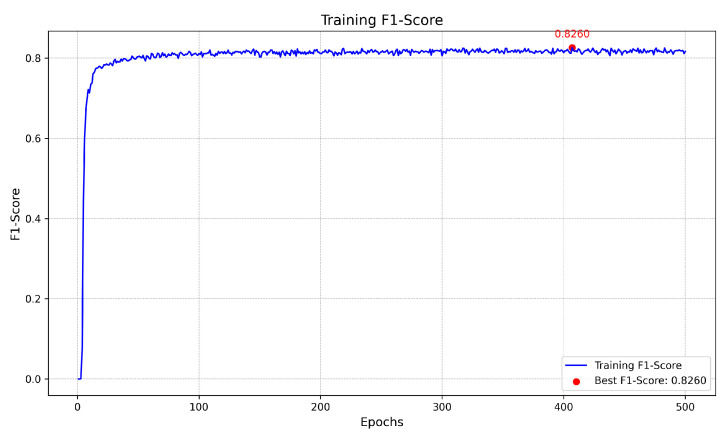
Meta-model training F1-Score over 500 epochs with best value annotated.

**Figure 16 biomedicines-13-00141-f016:**
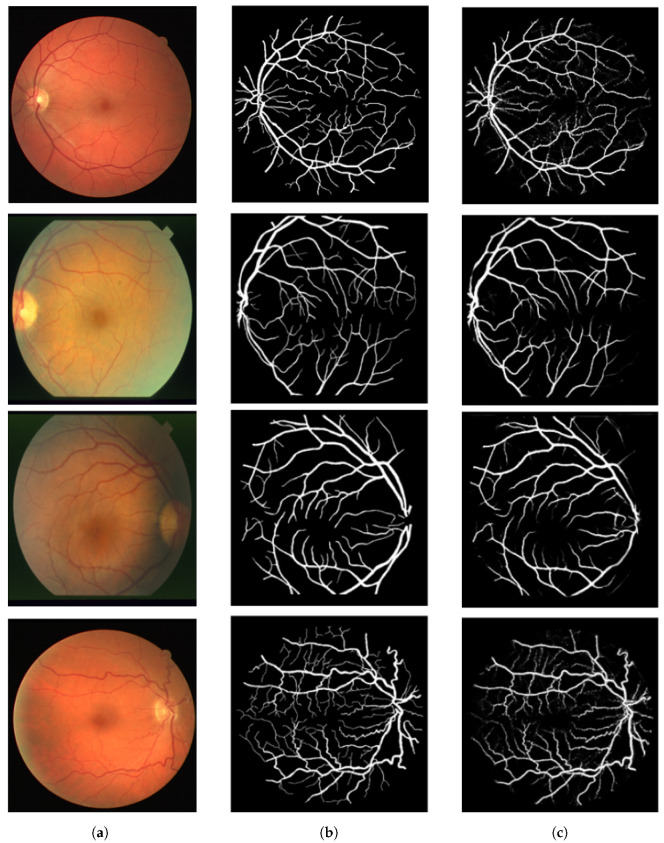
Meta-model predictions for retinal blood vessel segmentation on DRIVE and STARE datasets. (**a**) Original image, (**b**) ground truth, (**c**) meta-model prediction.

**Figure 17 biomedicines-13-00141-f017:**
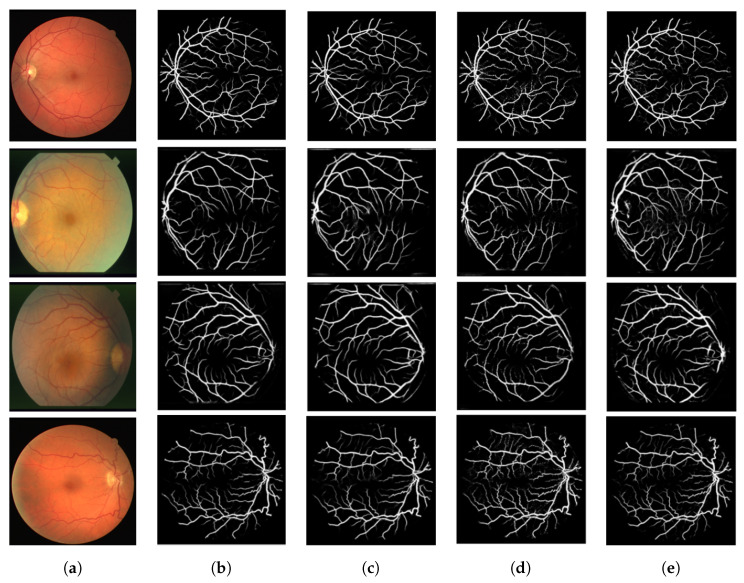
Individual model predictions for retinal blood vessel segmentation on DRIVE and STARE datasets. (**a**) Original image, (**b**) U-Net, (**c**) ResNet50, (**d**) CTU-Net, (**e**) U-Net with ResNet50 backbone.

**Table 1 biomedicines-13-00141-t001:** Performance comparison of meta-model combinations for blood vessel segmentation.

Meta-Model	Loss	Acc	SN	SP	AUC	F1-Score
U-Net + ResNet50	0.0730	0.9732	0.7787	0.9890	0.9829	0.8130
ResNet50 + CTU-Net	0.0757	0.9728	0.7821	0.9889	0.9835	0.8093
ResNet50 + U-Net-ResNet50-backbone	0.0740	0.9730	0.7681	0.9897	0.9812	0.8101
U-Net + CTU-Net	0.0781	0.9721	0.7770	0.9878	0.9805	0.8037
U-Net + U-Net-ResNet50-backbone	0.0749	0.9724	0.7631	0.9894	0.9827	0.8053
ResNet50 + U-Net + CTU-Net	0.0725	0.9736	0.7631	0.9894	0.9853	0.8141
ResNet50 + U-Net + U-Net-ResNet50-backbone	0.0698	0.9736	0.7836	0.9890	0.9835	0.8165
ResNet50 + CTU-Net + U-Net-ResNet50-backbone	0.0706	0.9737	0.7838	0.9891	0.9850	0.8172
U-Net + CTU-Net + U-Net-ResNet50-backbone	0.0721	0.9732	0.7862	0.9883	0.9849	0.8143
All architectures	**0.0591**	**0.9778**	**0.7790**	**0.9920**	**0.9912**	**0.8231**

**Table 2 biomedicines-13-00141-t002:** Performance metrics for blood vessel segmentation for the four proposed architectures and the meta-model on the DRIVE dataset.

Model	Loss	ACC	SN	SP	AUC	F1-Score
U-Net	0.0698	0.9739	0.7512	0.9879	0.9851	0.7933
ResNet50	0.0704	0.9756	**0.8039**	0.9877	0.9857	0.8129
U-Net-ResNet50-backbone	0.0706	0.9743	0.7605	0.9895	0.9845	0.7965
CTU-Net	0.0684	0.9752	0.7729	0.9895	0.9880	0.8047
Meta-model	**0.0591**	**0.9778**	0.7790	**0.9920**	**0.9912**	**0.8231**

**Table 3 biomedicines-13-00141-t003:** Performance metrics for blood vessel segmentation for the four proposed architectures and the meta-model on the STARE dataset.

Model	Loss	ACC	SN	SP	AUC	F1-Score
U-Net	0.0964	0.9663	0.8167	0.9769	0.9804	0.7618
ResNet50	0.0735	0.9736	0.7624	0.9885	0.9853	0.7921
U-Net-ResNet50-backbone	0.0850	0.9683	0.6409	**0.9914**	0.9778	0.7275
CTU-Net	0.0942	0.9689	**0.8497**	0.9773	0.9843	0.7828
Meta-model	**0.0649**	**0.9760**	0.7630	0.9912	**0.9855**	**0.8082**

**Table 4 biomedicines-13-00141-t004:** Quantitative results of our proposed approach in comparison to other deep learning methods on DRIVE dataset.

Study	ACC	SN	SP	AUC	F1-Score
Getahun et al. [18]	0.9702	**0.8421**	0.9828	0.9884	**0.8314**
Desiani [27]	0.9636	0.7871	0.9810	-	0.7890
Sanjeewani et al. [11]	0.9577	0.7436	0.9838	-	0.7931
Kim et al. [28]	-	-	0.9826	-	0.7840
Bhuiya et al. [29]	0.9698	0.8250	0.9837	0.9825	0.8270
Khan [30]	0.9698	0.8250	0.9837	0.9825	0.8270
Yin et al. [31]	0.9578	0.8038	0.9837	0.9846	-
Ibtehaz and Rahman [32]	0.9677	0.7928	0.9845	0.9677	-
Yan et al. [33]	0.9542	0.7653	0.9818	0.9752	-
Cao et al. [34]	0.9571	0.8123	0.9781	0.9809	-
Ye et al. [35]	0.9616	0.8268	0.9847	0.9864	0.8427
**Our**	**0.9778**	0.7790	**0.9920**	**0.9902**	0.8231

**Table 5 biomedicines-13-00141-t005:** Quantitative results of our proposed approach in comparison to other deep learning methods on STARE dataset.

Study	ACC	SN	SP	AUC	F1-Score
Getahun et al. [18]	0.9750	0.8393	0.9852	0.9916	0.8261
Desiani et al. [27]	0.9639	0.7935	0.9852	-	0.7716
Kim et al. [28]	-	-	0.9810	-	0.8189
Bhuiya et al. [29]	0.9747	0.8190	0.9874	0.9706	**0.8286**
Wu et al. [36]	0.9661	0.8132	0.9814	0.9860	-
Khan et al. [30]	0.9659	**0.8397**	0.9792	0.9810	-
Ibtehaz et al. [32]	0.9670	0.8239	0.9813	0.9871	-
Cao et al. [34]	0.9640	0.8539	0.9769	0.9877	-
Ye et al. [35]	0.9708	0.8936	0.9847	**0.9921**	0.8785
**Our**	**0.9760**	0.7630	**0.9912**	0.9855	0.8081

## Data Availability

The data used in this work come mainly from public datasets (see Section 4.1).

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
