# Peer review of "A Novel Ensemble Meta-Model for Enhanced Retinal Blood Vessel Segmentation Using Deep Learning Architectures"

_biomedicines, 2025, doi:10.3390/biomedicines13010141_

Round 1
Reviewer 1 Report
Comments and Suggestions for Authors
This paper presents a novel ensemble learning framework that integrates four deep learning architectures. Every architecture is tailored to improve feature extraction and segmentation efficacy. The results illustrate the effectiveness of the proposed technique in detecting thin retinal blood vessels. The comparative research of qualitative and quantitative outcomes with distinct models underscores the resilience of the ensemble structure, particularly in scenarios characterized by noise and diminished visibility. Should those issues be addressed, this research possesses the possibility for publication in the MDPI Journal of Biomedicines:

Author Response
Thank you for taking the time to review our paper, here is a point-by-point response to your comments and concerns.

Reviewer 2 Report
Comments and Suggestions for Authors
See the attachment.

Author Response

(The authors gave the same response as above.)

Reviewer 3 Report
Comments and Suggestions for Authors
This manuscript presents a novel approach to retinal blood vessel segmentation by leveraging an ensemble meta-model integrating multiple deep learning architectures. The study emphasizes the clinical importance of accurate retinal blood vessel segmentation for diagnosing diseases such as diabetic retinopathy, glaucoma, and hypertensive retinopathy. The authors propose a method aimed at improving segmentation accuracy by addressing challenges such as image noise, low contrast, and variability in vessel structure. The paper highlights its contributions by combining different deep learning models and demonstrating superior performance against benchmark datasets.
1. Clearly specify the unique contributions of the proposed method compared to existing ensemble approaches. Add a comparative table or discussion highlighting the differences.
2. Include a comprehensive description of the experimental setup. Justify the choice of metrics and explain why they are appropriate for evaluating segmentation performance.
3. Add qualitative results (e.g., segmented images) showing how the proposed method performs on various challenging scenarios, such as low-contrast images or noisy data.
4. Simplify the technical descriptions where possible and provide a brief introduction to key concepts, such as ensemble learning and deep learning architectures, for broader accessibility.
Author Response

(The authors gave the same response as above.)

Round 2
Reviewer 1 Report
Comments and Suggestions for Authors
Most of the replies satisfy me, but the author information in Refs. 7-9 were lost. Please modify them.
Author Response
We sincerely appreciate your thoughtful feedback and the time you have dedicated to reviewing our manuscript. Your comments have greatly contributed to improving the quality of our work.
1- Most of the replies satisfy me, but the author information in Refs. 7-9 were lost. Please modify them.
Thank you for your valuable feedback. The missing author information in References 7–9 has been carefully reviewed and corrected as per your suggestion.
Reviewer 2 Report
Comments and Suggestions for Authors
The author has completed the previous proposed changes. It is recommended that it can be accepted and published in current version.
Author Response
1-The author has completed the previous proposed changes. It is recommended that it can be accepted and published in current version.
Thank you for your positive feedback and recommendation for acceptance. We appreciate your time and effort in reviewing our manuscript and providing constructive comments that helped improve its quality.
Reviewer 3 Report
Comments and Suggestions for Authors
The authors addressed the concerns. The MS can be accepted now.
Author Response
1-The authors addressed the concerns. The MS can be accepted now.
Thank you for your positive assessment and recommendation for acceptance. We truly appreciate your constructive feedback and the time you dedicated to reviewing our manuscript.